# Calcium-Dependent Protein Kinase GhCDPK16 Exerts a Positive Regulatory Role in Enhancing Drought Tolerance in Cotton

**DOI:** 10.3390/ijms25158308

**Published:** 2024-07-30

**Authors:** Mengyuan Yan, Meijie Chai, Libei Li, Zhiwei Dong, Hongmiao Jin, Ming Tan, Ziwei Ye, Shuxun Yu, Zhen Feng

**Affiliations:** The Key Laboratory for Quality Improvement of Agricultural Products of Zhejiang Province, College of Advanced Agricultural Sciences, Zhejiang A&F University, Hangzhou 311300, China; yanmengyuan@zafu.edu.cn (M.Y.); 2022101012002@stu.zafu.edu.cn (M.C.); libeili@zafu.edu.cn (L.L.); 2021101011013@stu.zafu.edu.cn (Z.D.); jhm@stu.zafu.edu.cn (H.J.); 2023101011012@stu.zafu.edu.cn (M.T.); winlea@stu.zafu.edu.cn (Z.Y.)

**Keywords:** *GhCDPK16*, *Gossypium hirsutum* L., drought stress, osmotic adjustment, reactive oxygen species (ROS)

## Abstract

Cotton is essential for the textile industry as a primary source of natural fibers. However, environmental factors like drought present significant challenges to its cultivation, adversely affecting both production levels and fiber quality. Enhancing cotton’s drought resilience has the potential to reduce yield losses and support the growth of cotton farming. In this study, the cotton calcium-dependent protein kinase GhCDPK16 was characterized, and the transcription level of *GhCDPK16* was significantly upregulated under drought and various stress-related hormone treatments. Physiological analyses revealed that the overexpression of *GhCDPK16* improved drought stress resistance in Arabidopsis by enhancing osmotic adjustment capacity and boosting antioxidant enzyme activities. In contrast, silencing *GhCDPK16* in cotton resulted in increased dehydration compared with the control. Furthermore, reduced antioxidant enzyme activities and downregulation of ABA-related genes were observed in *GhCDPK16*-silenced plants. These findings not only enhanced our understanding of the biological functions of *GhCDPK16* and the mechanisms underlying drought stress resistance but also underscored the considerable potential of *GhCDPK16* in improving drought resilience in cotton.

## 1. Introduction

The escalating frequency and severity of droughts, exacerbated by global warming, pose profound challenges to agricultural systems worldwide. Cotton, as a primary source of natural fiber crucial to the textile industry worldwide, faces significant challenges from drought-induced stress. This environmental factor has led to substantial reductions in cotton production across major global regions, profoundly impacting market dynamics and economic stability [1]. Efforts to mitigate these challenges are pivotal, with research focusing increasingly on the genetic mechanisms underlying drought tolerance in cotton. Understanding these mechanisms not only promises to enhance our grasp of cotton’s adaptive responses to stress but also holds the key to developing resilient cultivars capable of withstanding future climatic uncertainties. Therefore, the exploration of more drought-related genes and their role in cotton is crucial for understanding how these genes contribute to the plant’s ability to withstand drought stresses and offer promising avenues for augmenting cotton’s resilience to drought, thereby safeguarding both the productivity and fiber quality of cotton.

Calcium (Ca^2+^) functions as a critical secondary messenger in plant cells, playing a pivotal role in regulating diverse physiological responses essential for adapting to environmental stresses [2]. Under stress conditions like drought and exposure to abscisic acid (ABA), the dynamic regulation of cytoplasmic Ca^2+^ levels initiates a series of molecular events crucial for plant survival and growth. Plant cells decode Ca^2+^ signals from diverse environmental and developmental cues through calcium-binding proteins to elicit transcriptional and metabolic responses [3,4]. This fundamental ion governs cellular processes through a complex network of calcium sensors, which include calmodulins (CaMs), calcineurin B-like (CBL) proteins, and calcium-dependent protein kinases (CDPKs) [5,6]. CDPKs are particularly noteworthy as they integrate a Ca^2+^-sensing domain with a protein kinase-activating domain within a single protein structure [7]. This unique feature enables CDPKs to act as sensor responders, binding Ca^2+^ directly and translating these signals into protein phosphorylation events. Moreover, CDPKs exhibit versatility in their ability to move between different subcellular compartments, facilitating a broad spectrum of cellular functions [8].

CDPKs play crucial roles in orchestrating plant responses to diverse abiotic stresses, including drought, salt, and cold. These proteins act as key integrators of signaling pathways that enhance stress tolerance by responding to signals from ABA, a pivotal hormone involved in plant adaptation to environmental challenges [9,10,11]. For instance, AtCPK4 and AtCPK11 phosphorylate ARF1 and ARF4, contributing to ABA signaling pathways during salt and drought stress [12]. Similarly, AtCDPK10 in Arabidopsis regulates ABA-mediated stomatal movement, thereby enhancing drought tolerance [13]. Additionally, AtCDPK8 maintains H_2_O_2_ homeostasis and regulates stomatal movement in response to fluctuations in cytosolic Ca^2+^ levels, underscoring the multifaceted roles of CDPKs in stress responses [14]. Moreover, AtCDPK13 modulates stomatal regulation under light-induced stress, further highlighting its role in fine-tuning plant responses to varying environmental cues [15]. Furthermore, CDPKs such as AtCPK6, AtCPK3, AtCPK21, and AtCPK23 interact with and phosphorylate SLAC1, an ABA-regulated S-type anion channel protein that promotes stomatal closure by facilitating anion efflux under stress conditions [16,17,18,19,20]. In rice, OsCDPK9 regulates stomatal aperture in response to ABA, thereby enhancing water retention and conferring drought tolerance [21]. Similarly, in maize, the homologous anion channel ZmSLAC1 is activated by ZmCDPK35 and ZmCDPK37, facilitating anion efflux to mediate stomatal closure under drought stress conditions [22]. However, comprehensive investigations into their specific roles in drought stress tolerance in cotton remain limited, presenting a significant gap in current understanding.

The *CDPK* family genes have been comprehensively characterized within the upland cotton genome, with multiple members expressed across diverse tissues, including roots, stems, leaves, petals, stamens, flowers, ovules, and fibers, spanning various developmental stages [23,24,25]. This broad expression profile indicates their potential diverse functional roles within the cotton. Specifically, GhCDPK1 has been reported to be involved in the calcium signaling events related to fiber elongation by phosphorylating GhACS2 [26,27]. Furthermore, silencing the *GhCDPK28-6* gene in cotton has been demonstrated to enhance plant resistance to Verticillium wilt by increasing reactive oxygen species (ROS), lignin, and callose accumulation [25]. Our previous research showed that GhCDPK60 enhanced drought stress tolerance by enhancing osmotic adjustment and mitigating ROS accumulation in plants [23]. Additionally, GhCDPK84 and GhCDPK93 regulated cotton fiber elongation through the phosphorylation of GhSUS2 at the residue Ser11 during fiber development [28]. Despite the pivotal roles of CDPKs in cotton’s response to various abiotic stresses, research focusing specifically on CDPKs in relation to drought stress in cotton remains limited. In this study, we investigated the potential role of GhCDPK16 in conferring drought tolerance in plants. *GhCDPK16* exhibited significantly increased expression levels in response to drought and various hormone treatments. To elucidate its function in regulating drought tolerance, we performed heterologous overexpression in Arabidopsis and virus-induced gene silencing (VIGS) in cotton. Our results indicated that GhCDPK16 positively regulated drought tolerance by facilitating osmotic adjustment, reducing ROS accumulation, and modulating the expression of drought stress-related genes, offering a promising candidate for genetic improvement of drought resistance in cotton.

## 2. Results

### 2.1. GhCDPK16 Possesses Typical Characteristics of CDPK Proteins

According to a previous study, AtCDPK4 and AtCDPK11 have roles in ABA-mediated drought tolerance [12]. Given the high similarity in protein sequences between GhCDPK16 and AtCDPK4/11 (Appendix A), it is speculated that GhCDPK16 may play a role in drought stress tolerance. *GhCDPK16* (*Ghir_A05G028110.1*) has an open reading frame of 1509 bp, encoding a 56.47 kDa protein. Through SMART analysis of the conserved domains of GhCDPK16 protein, it was found that GhCDPK16 protein contains an S_TKc (serine/threonine protein kinase) domain located at amino acids 31-289, which serves as the catalytic domain of GhCDPK16. Additionally, the GhCDPK16 protein also contains a domain comprising four EF-hand motifs spanning amino acids 336-470 (Figure 1A). Further, the crystal structure of GhCDPK16 was predicted using Alphafold3 (Figure 1B–D), and simulations of calcium ion binding states were conducted. The results demonstrated that each EF-hand consists of a 12-residue loop with an α-helix on each side capable of binding Ca^2+^ ions (Figure 1E). The above results indicated that GhCDPK16 possesses typical characteristics of the CDPK family proteins. Upon Ca^2+^ binding, GhCDPK16 undergoes conformational changes, revealing its active sites, thereby promoting the autophosphorylation and/or phosphorylation of downstream targets.

### 2.2. Expression of GhCDPK16 Is Enhanced in Response to Drought Stress and Hormonal Signals in Cotton

To elucidate the potential roles of GhCDPK16, we extracted a 2000 bp sequence upstream of the *GhCDPK16* start codon to gain deeper insights into GhCDPK16’s regulatory mechanisms. Analysis of the *GhCDPK16* gene promoter revealed several *cis*-regulatory elements associated with plant hormone responses and responses to biotic and abiotic stresses, including ABRE, ERE, MYB, MYC, STRE, and WUN-motif (Appendix A). Elements in the promoter have the capacity to regulate stress responses or the tissue-specific expression of *GhCDPK16* under diverse environmental conditions. Considering the close relationship between *cis*-acting elements and gene expression, we initially investigated the tissue-specific expression patterns of *GhCDPK16*. *GhCDPK16* exhibited expression in all tissues (Figure 2A), suggesting its involvement throughout cotton’s growth and development. Plant hormones play a pivotal role in regulating plant growth and survival under diverse environmental stresses. To verify *GhCDPK16*′s responsiveness to hormonal signals, we analyzed its expression profile under various exogenous hormone treatments. The results indicated that *GhCDPK16* expression peaked at 6 h post-treatment with abscisic acid (ABA), methyl jasmonate (MeJA), salicylic acid (SA), gibberellin (GA), and auxin (IAA), followed by a subsequent decline (Figure 2B–F). Moreover, exposure to simulated drought stress using PEG resulted in a sustained increase in *GhCDPK16* expression, reaching its highest level at 24 h (Figure 2G). These results indicated that *GhCDPK16* can respond to various biotic and abiotic stress-related hormones and may play a key role in cotton’s response to drought stress.

### 2.3. GhCDPK16 Is a Cytoplasmic and Nuclear Localization Protein

CDPKs were predicted to localize on multiple cellular compartments, including the endoplasmic reticulum, plasma membrane, and cytoplasm, as well as in the nucleus [11]. Most CDPKs contain both an N-myristoylation site and a palmitoylation site, which together determine their subcellular distribution [29]. We used GPS-Lipid 1.0 [30] to predict the N-myristoylation and palmitoylation sites of GhCDPK16, revealing that GhCDPK16 has only one palmitoylation site at the N-terminus. Previous studies have indicated that CDPK proteins lacking one or both N-acylation sites were primarily localized in the cytoplasm and nucleus [29]. To confirm the subcellular localization of GhCDPK16, GFP was fused to the N-terminus or C-terminus of the GhCDPK16 protein under the constitutive CaMV35S promoter. Transient expression was conducted in tobacco leaves using Agrobacterium infiltration, with a free GFP vector used as a positive control (Figure 3). Results indicated that free GFP localized to the cell membrane, cytoplasm, and nucleus, while GFP-GhCDPK16 and GhCDPK16-GFP were both localized in the cytoplasm and nucleus (Figure 3). These findings demonstrated that the fusion of GFP to either the N-terminus or C-terminus of GhCDPK16 did not alter its subcellular localization and GhCDPK16 functions in the cytoplasm and nucleus.

### 2.4. The Overexpression of GhCDPK16 Improves Drought Tolerance in Arabidopsis

To investigate the function of GhCDPK16 under drought stress, *GhCDPK16* was overexpressed in wild-type Arabidopsis. Three independent homozygous lines (OE1, OE2, and OE3) from the T_3_ generation were selected for subsequent experiments. RT-qPCR analysis showed a significant increase in *GhCDPK16* gene expression in OE1, OE2, and OE3 plants compared with the wild-type ones (Appendix A). To assess the impact of GhCDPK16 on drought tolerance in transgenic Arabidopsis, seeds from wild-type and *GhCDPK16*-overexpressing lines were germinated on 1/2 MS medium supplemented with 0 mM, 100 mM, 200 mM, or 300 mM mannitol, and their germination situation was observed after one week (Figure 4A). The results showed that on mannitol-free (0 mM) medium, both wild-type and *GhCDPK16*-overexpressing lines germinated normally and displayed consistent growth. However, at 100 mM mannitol, the growth of *GhCDPK16*-overexpressing lines was slightly superior to that of the wild-type ones. With increasing mannitol concentrations, the germination situation of *GhCDPK16*-overexpressing lines significantly surpassed that of the wild-type ones (Figure 4A). To further assess the impact of *GhCDPK16* overexpression on drought resistance in transgenic Arabidopsis seedlings, wild-type and overexpressing seedlings grown uniformly for 4 days were transferred to media with different mannitol concentrations (0 mM, 100 mM, 200 mM) for vertical growth for an additional 10 days. On control plates without mannitol treatment, there was no significant difference in root length among WT, OE1, OE2, and OE3 (Figure 4B). However, on media containing 100 mM and 200 mM mannitol, the root length of *GhCDPK16*-overexpressing seedlings was significantly greater than that of the wild-type ones, indicating improved growth performance (Figure 4C,D). Overall, these results indicated that the overexpression of *GhCDPK16* can enhance the germination vigor and rate of transgenic Arabidopsis under mannitol stress, as well as increase the tolerance of Arabidopsis seedlings to drought stress.

### 2.5. GhCDPK16 Overexpression Enhanced Antioxidant Capacity and Water Retention Ability

Plants have developed intricate morphological, physiological, and molecular mechanisms to effectively counteract or alleviate growth impairments triggered by drought stress, encompassing osmotic regulation to maintain cellular water balance, ensuring high relative water content (RWC) crucial for sustained metabolic activity, and the activation of stress-responsive genes that bolster resilience against drought-induced damage [31]. To clarify the physiological mechanisms underlying the drought resistance conferred by GhCDPK16 in plants, we measured and analyzed various physiological parameters in *GhCDPK16*-overexpressing lines under drought stress, including RWC, proline content, malondialdehyde (MDA) content and the activities of superoxide dismutase (SOD), peroxidase (POD) and catalase (CAT). Under normal conditions, *GhCDPK16*-overexpressing lines showed no significant differences compared with the WT ones in these physiological parameters (Figure 5A–F). However, following drought stress, the decrease in RWC was less pronounced in *GhCDPK16*-overexpressing lines compared with the WT ones (Figure 5A). Proline content increased under drought conditions in *GhCDPK16*-overexpressing lines compared with the WT ones (Figure 5B). Meanwhile, MDA levels were significantly lower in *GhCDPK16*-overexpressing lines than in the WT ones (Figure 5C), and the activities of SOD, POD, and CAT were significantly higher in *GhCDPK16*-overexpressing lines than in the WT ones (Figure 5D–F). These changes in physiological and biochemical parameters indicated that *GhCDPK16* overexpression enhanced the antioxidant capacity and water retention ability of transgenic Arabidopsis, thereby improving its adaptation to drought stress.

In prior research, the elevation of transcript levels of stress-responsive genes such as *AtDREB2A*, *AtNXH1*, *AtRD29A/B*, and *AtDi19* has been shown to significantly enhance the tolerance to drought stresses [32,33,34,35]. We assessed the transcript levels of these stress-responsive genes in *GhCDPK16*-overexpressing lines under drought conditions, revealing significantly higher levels of *AtDREB2A*, *AtNXH1*, *AtRD29A/B*, and *AtDi19* compared with controls (Appendix A). These results indicated that GhCDPK16 was involved in upregulating stress-related gene transcription in Arabidopsis, thereby enhancing its tolerance to drought stress.

### 2.6. Silencing GhCDPK16 Reduced Cotton Drought Tolerance by Affecting the ABA Signaling Pathway

To further validate whether GhCDPK16 is involved in the response to drought stress in cotton, the transcription level of *GhCDPK16* was lowered in cotton plants using virus-induced gene silencing (VIGS) technology, and the drought tolerance of *GhCDPK16*-silenced plants was investigated. In the VIGS experiment, plants injected with the TRV:*GhPDS* vector were used as a positive control, and the silencing efficiency of *GhCDPK16* was analyzed after the positive control plants exhibited the white symptomatic phenotype (Figure 6A). *GhCDPK16* gene expression was examined in three randomly selected TRV:00 and TRV:*GhCDPK16* plants, revealing significantly lower expression levels in TRV:*GhCDPK16* plants compared with the TRV:00 ones (Figure 6B). These results indicate the successful silencing of the *GhCDPK16* gene in upland cotton plants. To evaluate the performance of *GhCDPK16*-silenced plants under drought stress, TRV:*GhCDPK16* and control lines were subjected to a 14-day dewatering treatment. The drought treatment results showed that TRV:*GhCDPK16* plants exhibited more severe wilting and yellowing of leaves compared with TRV:00 plants (Figure 6C). Subsequently, physiological parameters were measured in TRV:00 and TRV:*GhCDPK16* plants after the treatment. Chlorophyll, RWC, and proline content in TRV:*GhCDPK16* leaves was significantly lower compared with TRV:00 plants under drought conditions (Figure 6D–F), indicating more severe osmotic stress in the plants after *GhCDPK16* silencing. In drought conditions, MDA content in TRV:*GhCDPK16* plant leaves was significantly higher than in TRV:00 plants (Figure 6G), indicating more severe oxidative damage. In addition, the activities of antioxidant enzymes such as SOD, POD, and CAT were measured under drought stress and normal conditions. Under drought conditions, the activities of SOD, POD, and CAT in TRV:*GhCDPK16* plant leaves were significantly reduced (Figure 6H–J), suggesting impaired ROS scavenging capacity. These results collectively indicated that *GhCDPK16* gene silencing rendered cotton more susceptible to drought stress, possibly due to disrupted osmotic regulation and impaired antioxidant defense mechanisms.

ABA is pivotal in plant signal transduction pathways responding to drought stress [9,36]. Given that *GhCDPK16* was responsive to ABA signaling, we further investigated the expression levels of ABA-related genes under drought conditions to elucidate the molecular mechanisms underlying the reduced drought tolerance in *GhCDPK16*-silenced plants. Genes involved in ABA biosynthesis pathways (*GhABA1*, *GhABA2*, and *GhAAO3*) exhibited significantly decreased expression in TRV:*GhCDPK16* plants compared with TRV:00 plants (Figure 7A–C). The expression of *GhCDPK1*, a known positive regulator induced by drought stress, was notably diminished in TRV:*GhCDPK16* lines relative to TRV:00 ones (Figure 7D). Additionally, *GhDi19*, a zinc finger transcription factor recognized for its role in promoting drought responses and ABA signaling, showed significantly reduced expression in TRV:*GhCDPK16* plants compared with TRV:00 plants under drought conditions (Figure 7E). These findings suggest that GhCDPK16 likely enhances cotton’s drought tolerance by positively regulating ABA signaling and modulating the expression of drought-responsive transcription factors.

## 3. Discussion

Cotton plays a key role in both industrial and agricultural economies as a vital commercial crop. However, the scarcity of water resources, ecological challenges such as soil salinity and extreme temperatures, environmental pollution, and various human activities have worsened the severity of land disputes related to food and cotton on a global scale [1,36,37]. Drought stress poses a significant abiotic threat to plants, inducing modifications in transpiration rates, impacting the development and composition of the photosynthetic apparatus, leading to excessive production of ROS, altering the biochemical composition, and ultimately impacting plant growth and reducing crop yield. In response, plants have evolved a complex network of signal transduction pathways to regulate metabolism and adapt to environmental conditions, such as the Ca^2+^ pathway mediated by CDPKs [11]. Previous studies have shown that homologous genes typically share similar biological functions. In this study, phylogenetic analysis revealed that the upland cotton gene *GhCDPK16* shared the highest homology with Arabidopsis genes *AtCDPK4* and *AtCDPK11*. These two Arabidopsis genes were known as critical positive regulators in the Ca^2+^-mediated ABA signaling pathway, influencing seed germination, seedling growth, and stomatal movement through the phosphorylation of downstream transcription factors ABF1 and ABF4, thereby enhancing stress tolerance [12]. Building upon these insights into the functions of Arabidopsis homologs, we examined the expression patterns of *GhCDPK16* under various hormone and osmotic stress treatments. Exposing normal cotton seedlings to exogenous ABA resulted in a significant increase in *GhCDPK16* expression after 6 h, demonstrating its responsiveness to ABA. Additionally, subjecting cotton seedlings to PEG-induced drought stress showed a consistent upward trend in *GhCDPK16* gene expression over time, suggesting its potential role in coordinating cotton’s response to drought stress through the ABA signaling pathway.

Plant CDPKs have been shown to exhibit diverse subcellular localizations, with the N-terminal region playing a crucial role [38]. N-terminal myristoylation and palmitoylation facilitate membrane localization, while those lacking these modifications are evenly distributed in the cytoplasm and nucleus [29]. Our subcellular localization results indicated that GhCDPK16 was found in both the nucleus and cytoplasm, underscoring its functional significance independent of N-terminal acylation. To investigate GhCDPK16’s involvement in plant drought response, we employed transgenic techniques to heterologously overexpress *GhCDPK16* in Arabidopsis. These transgenic lines were subjected to drought stress induced by mannitol, revealing significant alterations compared with wild-type Arabidopsis. Notably, the overexpression of *GhCDPK16* under drought conditions demonstrated improved germination rates, as well as the development of longer and more robust roots in seedlings compared with the wild-type ones. To further investigate GhCDPK16’s impact on cotton’s response to drought stress, we employed VIGS technology to silence *GhCDPK16* expression in cotton. Drought-treated silenced plants exhibited greater wilting and yellowing compared with controls. These results suggested that GhCDPK16 was a positive regulator of the response to drought stress.

Under limited water conditions, plants tend to accumulate compatible osmolytes like proline to lower cellular osmotic potential [39]. Proline is known for its pivotal role in plant responses to stressful environmental conditions by facilitating osmotic adjustments [40], mitigating ROS effects through antioxidant system stabilization, and safeguarding cell membrane integrity [40,41,42]. Moreover, proline is recognized for its role in upregulating genes associated with drought tolerance [43,44]. Our investigation into the physiological mechanisms further unveiled that GhCDPK16 regulated osmolyte proline content. MDA is a reliable diagnostic indicator for assessing the extent of injury in stressed plants [45]. Elevated levels of MDA in *GhCDPK16*-silenced lines underscored increased oxidative damage, reflective of severe cellular oxidative stress under drought conditions. Increasing evidence suggests that drought triggers the generation of ROS [46]. The role of antioxidant enzymes in ROS scavenging is pivotal for plant stress responses, such as CAT, SOD, and POD [47,48]. In our study, we observed that *GhCDPK16*-overexpressing Arabidopsis exhibited significantly increased activities of SOD, POD, and CAT enzymes, while *GhCDPK16*-silenced cotton showed markedly decreased activities of these enzymes, indicating that GhCDPK16 maintained ROS homeostasis crucial for drought stress adaptation by enhancing antioxidant enzyme activity.

To further comprehend the molecular mechanism of GhCDPK16 under drought stress, we conducted an analysis of the transcription levels of various stress-inducible genes. The downregulated expression of the stress-related transcription factor *GhDi19* and genes involved in ABA biosynthesis pathways (*GhABA1*, *GhABA2*, *GhAAO3*) in silenced plants suggest that *GhCDPK16* regulates the expression of genes related to the ABA signaling pathway during water stress in cotton. In conclusion, our findings offered preliminary evidence that GhCDPK16 positively enhances drought tolerance in cotton by modulating the ABA signaling pathway, balancing osmotic substance accumulation, and coordinating mechanisms for ROS scavenging.

## 4. Materials and Methods

### 4.1. Plant Materials and Growth Conditions

The upland cotton cultivar used in this study for subsequent expression analysis and VIGS experiments was the ‘TM-1’. *Arabidopsis thaliana* Columbia-0 type was employed for the heterologous overexpression analysis of the gene. *Nicotiana benthamiana* RA-4 accession was used as the transient transformation receptor for subcellular localization analysis. All plant materials were maintained by our laboratory. They were grown in the growth chambers (PRG-I, Ningbo Le Electrical Instrument Manufacturing Co., Ltd., Ningbo, China) with a constant temperature of 23 °C, 70% humidity, LED lighting with a light intensity of 500 µmol/m^2^·s, and a 16-h light and 8-h dark cycle.

### 4.2. Gene and Protein Sequence Analysis of GhCDPK16

The 2000 bp upstream sequence of the *GhCDPK16* promoter was retrieved and predicted using the online website plantCARE tool (http://bioinformatics.psb.ugent.be/webtools/plantcare/html/, accessed on 5 February 2023) to identify potential cis-regulatory elements. The theoretical isoelectric point, molecular weight, and instability coefficient of the GhCDPK16 protein were estimated using the ProParam website (https://web.expasy.org/protparam/, accessed on 5 February 2023). Conservative domain prediction was conducted using the SMART database (http://smart.embl-heidelberg.de/, accessed on 5 February 2023). Protein sequences of GhCDPK16 were loaded on Alphafold3.0 (https://alphafoldserver.com/, accessed on 5 February 2023) for the three-dimensional structure prediction. The structure and calcium binding states of proteins were visualized using PyMOL (v2.6.0a0).

### 4.3. Expression Analysis of GhCDPK16

To investigate the spatial and temporal expression characteristics of *GhCDPK16*, we extracted RNA from various tissues of ‘TM-1’ grown in the field (Hangzhou, Zhejiang Province, China), including root, stem, leaf, cotyledon, apical bud, anther, filament, stigma, style, petal, sepal, torus, and flower, as well as ovule and fiber at different developmental stages, respectively. The expression levels of *GhCDPK16* in these samples were then analyzed using qRT-PCR. To analyze the response of *GhCDPK16* to different hormones, three-leaf-stage cotton seedlings were evenly sprayed with 100 µM of each hormone (ABA, MeJA, SA, GA, IAA) for 24 h. To investigate the response of *GhCDPK16* to drought stress, three-leaf-stage cotton seedlings were subjected to root irrigation with a 15% PEG6000 solution. Leaf samples were collected at 0 h, 6 h, 12h, and 24 h post-treatment, respectively. RNA was extracted from these samples, and the expression levels of *GhCDPK16* were analyzed at each time point using qRT-PCR. Detailed procedures for RNA extraction and qRT-PCR can be found in previous reports [23,49,50,51], with *GhACT4* serving as an internal reference control [52]. All primers used in this study are listed in Appendix A.

### 4.4. Subcellular Localization Analysis

The coding sequence of *GhCDPK16*, with a length of 1509 bp, was downloaded from the COTTONGEN database (https://www.cottongen.org, accessed on 5 February 2023). Specific primers were designed using Primer Premier 5.0 for cloning the *GhCDPK16* and are listed in Appendix A. To analyze the subcellular localization of GhCDPK16, the full-length coding sequence of *GhCDPK16* was cloned and inserted into the pCAMBIA1305-GFP vector under the control of the *CaMY35S* promoter to generate GFP-fusion proteins, GhCDPK16-GFP, and GFP-GhCDPK16. The resulting plasmids were then transformed into the *Agrobacterium tumefaciens* strain GV3101 and infiltrated into the epidermal cells of 3–4-week-old *Nicotiana benthamiana* leaves. GFP signals were observed 48 h post-infiltration using a laser scanning confocal microscope (LSM 880, Zeiss, Oberkochen, Germany).

### 4.5. VIGS Assay

To investigate the phenotypes of *GhCDPK16* gene silencing under drought treatment in cotton, we targeted a 244 bp-specific region for *GhCDPK16* silencing and designed primers for amplification before integration into the TRV2 vector. Specific primers are shown in Appendix A. The recombinant plasmid was transformed into the GV3101 *Agrobacterium tumefaciens* strain and stored at −80 °C. The Agrobacterium containing different TRV2 recombinant vectors, including the empty vector TRV2 (negative control), the positive control vector TRV2:*GhPDS*, and the recombinant vector TRV2:*GhCDPK16*, were mixed with Agrobacterium containing the helper vector TRV1 at a ratio of 1:1, respectively. The mixed bacterial solution was used to infect the cotyledons of cotton seedlings. Following 10–14 days of growth in a growth chamber, the albino phenotype of the positive control was monitored, and qRT-PCR was employed to assess gene silencing in the TRV2:*GhCDPK16* lines. For drought treatment, when TRV2 and TRV2:*GhCDPK16* seedlings reached the three-leaf stage, they received uniform watering before undergoing water deficit treatment. After a 14-day treatment period, the phenotypes of the plants were captured, and samples were collected from various lines to assess physiological indicators and quantify the expression levels of genes related to drought stress. The *GhUBQ7* was used as an internal reference control [53].

### 4.6. Generation and Analysis of Transgenic Arabidopsis

To analyze the phenotype of *GhCDPK16* overexpression in plants under drought stress, the coding sequence of *GhCDPK16* was cloned into the pBI121 vector and transformed into *Agrobacterium tumefaciens* GV3101. The transformed bacteria carrying the correct plasmid were then used for the genetic transformation of Arabidopsis via the floral dip method [54]. Positive plants were selected on 1/2 MS solid medium [prepared by dissolving 2.47 g of 1/2 MS medium (HB8469-12, HopeBio, Qingdao, China), 10 g sucrose (10021463, SCR, Shanghai, China)—adjusting the pH to 5.7–5.8 with KOH—and 3 g Phytagel (P8169, Sigma, Darmstadt, Germany) in 1 L of water] appended with 50 mg/mL kanamycin until stable homozygous T_3_ generation lines were obtained, which were used for subsequent expression level assessment and phenotype observations. The gene expression of overexpression plants was quantified using qRT-PCR, with *AtSAND* serving as the internal reference gene [55].

To assess the drought tolerance of *GhCDPK16*-overexpressing plants during seed germination, seeds from both WT and *GhCDPK16*-overexpressing lines were surface-sterilized by soaking in 70% ethanol for 30 s, followed by immersion in 10% sodium hypochlorite for 5 min, and then washed five times with sterile water. After vernalization at 4 °C for 3 days, seeds were sown on 1/2 MS medium [prepared by dissolving 2.47 g of 1/2 MS medium, 10 g sucrose—adjusting the pH to 5.7–5.8 with KOH—and 3 g Phytagel in 1 L of water] in the plates (90 mm × 20 mm) supplemented with different concentrations of mannitol (100 mM, 200 mM, and 300 mM), with mannitol-free 1/2 MS medium used as a control. Plates were transferred to a growth chamber for germination, and photographs were taken at 7 days post-germination. Each line was represented by at least 80 seeds per treatment, with three replicates per experiment.

To examine the tolerance of *GhCDPK16*-overexpressing seedlings under drought stress, surface-sterilized and vernalized seeds of both WT and *GhCDPK16*-overexpressing lines were germinated on 1/2 MS medium for 4 days. Uniformly growing seedlings were then transferred to 1/2 MS medium [prepared by dissolving 2.47 g of 1/2 MS medium, 10 g sucrose—adjusting the pH to 5.7–5.8 with KOH—and 5 g Phytagel in 1 L of water] in the plates (10 cm × 10 cm × 1.5 cm) supplemented with 0 mM, 100 mM, or 200 mM mannitol for 10 days of osmotic stress treatment. We observed the phenotypic changes of the plants after stress treatment, measured root lengths and physiological parameters, and evaluated the differences in expression levels of drought-responsive genes.

### 4.7. Relative Water Content Assessment

For relative water content assessment, a predetermined quantity of plant leaves from *GhCDPK16*-overexpressing Arabidopsis or *GhCDPK16*-silenced cotton was initially collected, and their fresh weight (m1) was determined. Subsequently, the freshly weighed plant material was immersed in a tube containing 10 mL of water and allowed to equilibrate at 4 °C in a refrigerator for 24 h until fully saturated. Following removal, excess surface water was removed by blotting with filter paper, and the material was promptly reweighed to determine the saturated fresh weight (m2). The plant material with the determined saturated fresh weight was then enclosed in a drying envelope and subjected to brief blanching at 105 °C for 10 min, followed by drying at 80 °C until a constant weight was reached (m3). The formula for computing relative water content (RWC) was expressed as RWC (%) = (m1 − m3)/(m2 − m3) × 100% [56].

### 4.8. Determination of Chlorophyll Content

To measure chlorophyll content, 0.5 g of fresh leaves from *GhCDPK16*-silenced cotton were shredded and placed into a 25 mL flask for extraction using a 4.5:4.5:1 ethanol/acetone/distilled water solvent mixture. The flask was then wrapped in aluminum foil and stored at 4 °C for 2–3 days until the leaves became fully bleached. Following this, the chlorophyll acetone extract was quickly extracted, and its absorbance was measured at 645 nm and 663 nm using a spectrophotometer. Chlorophyll content (mg/g FW) was calculated using the formula (20.2 × A_645_ + 8.02 × A_663_) × V_T_/(W × 1000), where V_T_ is the extract volume and W is the leaf weight [57].

### 4.9. Determination of Proline Content

For proline content determination, 0.1 g of plant seedling leaves of *GhCDPK16*-overexpressing Arabidopsis or *GhCDPK16*-silenced cotton, both before and after drought treatment, were powdered and transferred to 2 mL tubes. A 1.5 mL aliquot of 3% sulfosalicylic acid solution was added, followed by heat extraction in a boiling water bath for 10 min. After cooling, samples were centrifuged at 12,000 rpm for 10 min, and 1 mL of supernatant was diluted to 10 mL with 3% sulfosalicylic acid. Subsequently, 2 mL of sample solution was mixed with 2 mL each of acetic acid and acidic ninhydrin in glass tubes. The mixture was incubated in a boiling water bath for 30 min, cooled, and then added to 5 mL of toluene solution. The proline concentration in the toluene fraction was quantified by measuring absorbance at 520 nm, referencing a toluene blank. This was compared against absorbance values of a series of standard L-proline solutions (0–20 μg/mL) prepared similarly to the plant samples. Proline content (μg/g FW) in plant materials was calculated using the formula (C × Vt)/(W × Vs), where C is the proline weight obtained from a standard curve (μg), Vt is the total extraction volume (mL), Vs is the volume of extraction solution used for measurement (mL), and W is the sample weight (g) [58,59].

### 4.10. Determination of MDA Content and Antioxidant Enzyme Activities

Fresh leaves (0.5 g) from *GhCDPK16*-overexpressing Arabidopsis or *GhCDPK16*-silenced cotton subjected to drought treatment, both before and after treatment, were ground in 8 mL of phosphate buffer on ice. The resulting mixture underwent centrifugation at 12,000 rpm for 25 min at 4 °C, yielding a supernatant (5.0–6.0 mL) used for assessing the activities of SOD, POD, and CAT. Further experimental details were available in prior studies [60,61]. For the measurement of MDA content, 1.5 mL of crude enzyme solution was combined with 2.25 mL of 6% (*w*/*v*) thiobarbituric acid. The mixture was then subjected to a 12-min incubation in a boiling water bath, followed by rapid cooling. Subsequently, the sample was centrifuged at 4000 rpm for 15 min, and the absorbance at 450 nm, 532 nm, and 600 nm in the resulting supernatant was measured. The calculation of MDA content follows the formula MDA content (μmol/g, FW) = [6.45 × (A_532_ − A_600_) − 0.56 × A_450_] × V_T_/(W × 1000), where V_T_ denotes the total volume of the extract in milliliters and W signifies the sample weight in grams [62]. All absorbances were recorded with an Absorbance Microplate Reader (SpectraMax^®^190, Molecular Devices, San Jose, CA, USA), and all experiments were performed with three biological replicates.

## 5. Conclusions

In this study, we analyzed the protein characteristics of GhCDPK16, which contains typical S_TKc protein kinase domains and EFh calcium-binding domains. Expression analysis revealed that *GhCDPK16* was expressed across various tissues of cotton and responded to various stress-related hormones and PEG-induced osmotic stress. GhCDPK16 was localized in both the cytoplasm and nucleus as a calcium-dependent protein kinase. The overexpression of *GhCDPK16* in Arabidopsis enhanced plant resistance to osmotic stress, whereas its silencing in cotton reduced antioxidant enzyme activity, downregulated ABA-related genes, and severely compromised drought tolerance. Our findings indicate that GhCDPK16 plays a regulatory role in drought resistance in plants, offering a new genetic resource for enhancing drought tolerance in cotton.

## Figures and Tables

**Figure 1 ijms-25-08308-f001:**
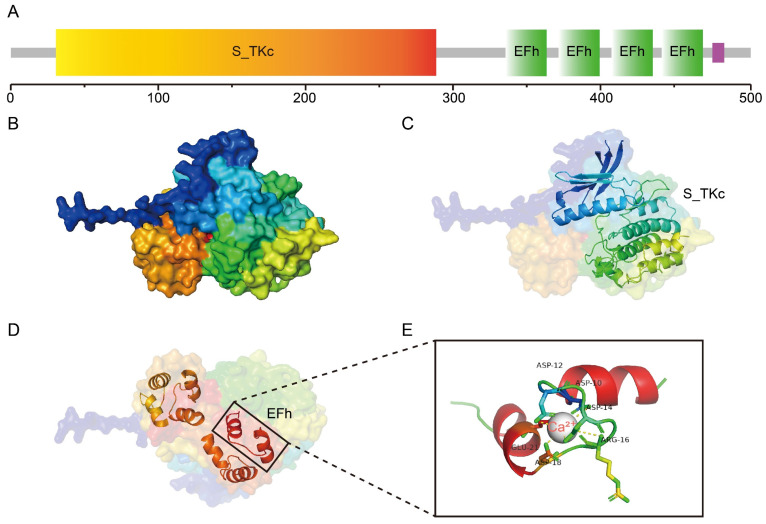
GhCDPK16 possesses typical characteristics of CDPK proteins. (**A**) Conservative domains of GhCDPK16 protein. (**B**) Model of the GhCDPK16 protein predicted by AlphaFold3. (**C**,**D**) Cartoon of S_TKc (**C**) and EFh (**D**) structures. (**E**) Docking of Ca^2+^ in the EFh domain of GhCDPK16.

**Figure 2 ijms-25-08308-f002:**
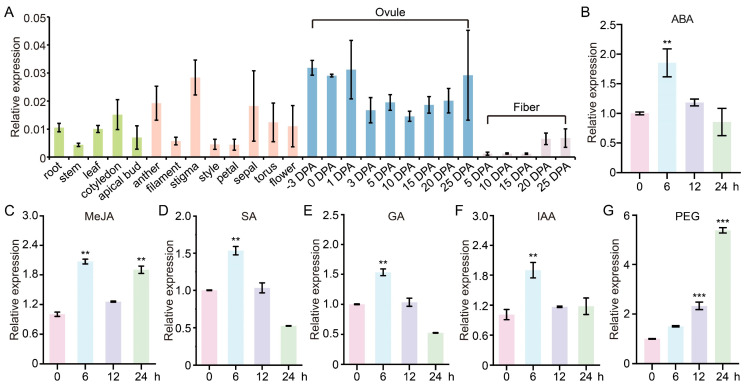
Expression analysis of *GhCDPK16* during cotton development and hormone treatment. (**A**) The expression profile of *GhCDPK16* in different tissues. (**B**–**F**) Transcription level of *GhCDPK16* in leaves of cotton after ABA (**B**), MeJA (**C**), SA (**D**), GA (**E**), and IAA (**F**) treatments. (**G**) Expression patterns of *GhCDPK16* under PEG-induced drought stress. Values represented the mean ± SD from three biological replicates. ** *p* < 0.01 and *** *p* < 0.001 by Student’s *t* test.

**Figure 3 ijms-25-08308-f003:**
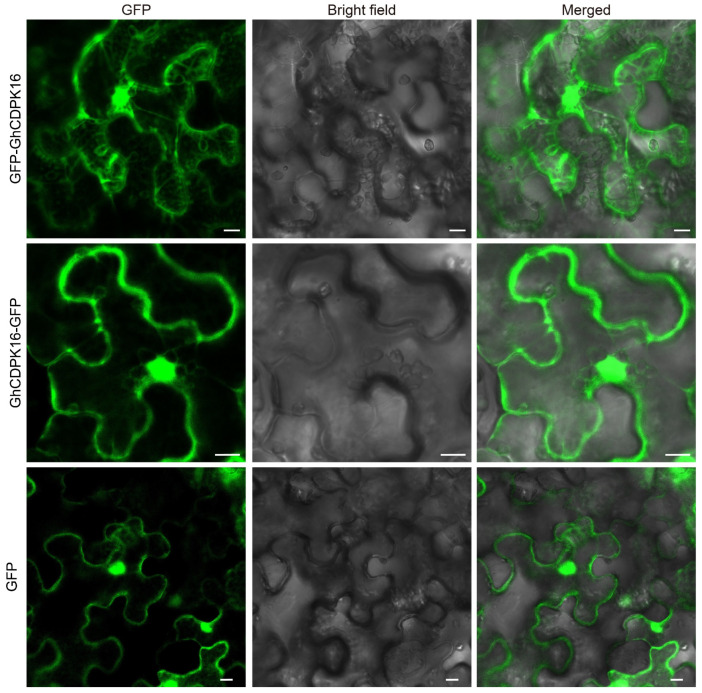
Subcellular localization of GhCDPK16. GFP-GhCDPK16 and GhCDPK16-GFP fusion proteins were transiently expressed in *N. benthamiana* leaf cells. The empty GFP protein served as the positive control. Bars = 10 μm.

**Figure 4 ijms-25-08308-f004:**
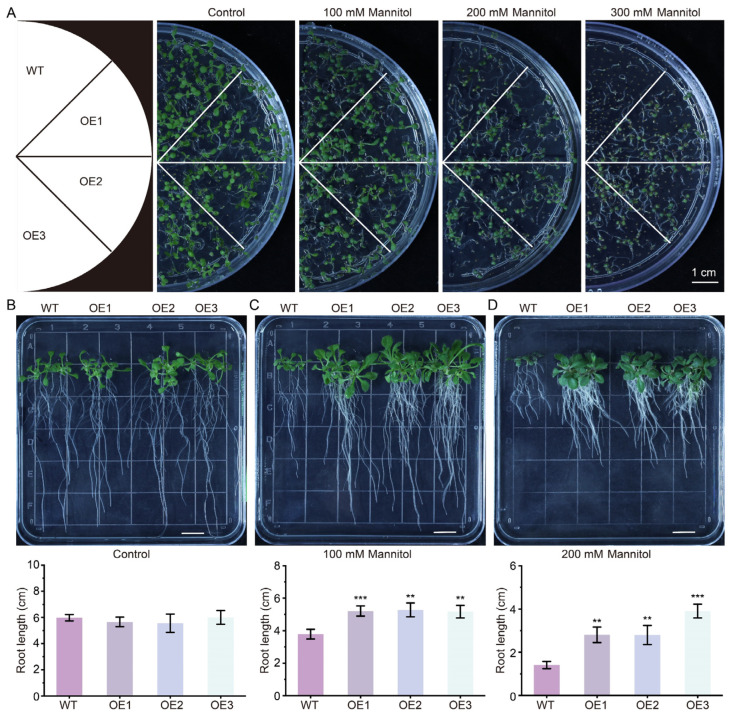
Overexpression of *GhCDPK16* increased drought resistance in Arabidopsis. (**A**) Seed germination of wild-type (WT) and *GhCDPK16*-overexpressing Arabidopsis on 1/2 MS agar plates with 0 mM, 100 mM, 200 mM, and 300 mM mannitol. Photographs were taken one week after mannitol treatments. OE1–OE3 represented independent homozygous Arabidopsis lines overexpressing *GhCDPK16*. Bars = 1 cm. (**B**–**D**) Phenotypes of WT and overexpressing lines after 10 days of treatment with 0 mM (**B**), 100 mM (**C**), and 200 mM (**D**) mannitol, and the statistics of primary root length under different concentration of mannitol treatments. Bars = 1 cm. Values represented the mean ± SD from three biological replicates. ** *p* < 0.01 and *** *p* < 0.001 by Student’s *t* test.

**Figure 5 ijms-25-08308-f005:**
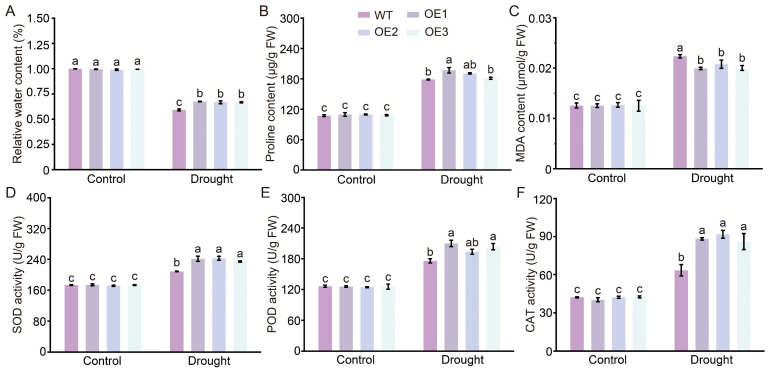
Overexpression of *GhCDPK16* enhanced antioxidant capacity and water retention ability. (**A**–**C**) The analysis of the relative water content (**A**), proline content (**B**), and MDA content (**C**) of WT and transgenic lines with or without drought treatment for 10 days. (**D**–**F**) Measurements of SOD (**D**), POD (**E**), and CAT (**F**) activities in leaves with WT and *GhCDPK16*-overexpressing lines after drought treatment. OE1–OE3 represented independent homozygous Arabidopsis lines overexpressing *GhCDPK16*. Values represented the mean ± SD from three biological replicates. Columns with different letters indicated significant differences (*p* < 0.05, Student’s *t* test).

**Figure 6 ijms-25-08308-f006:**
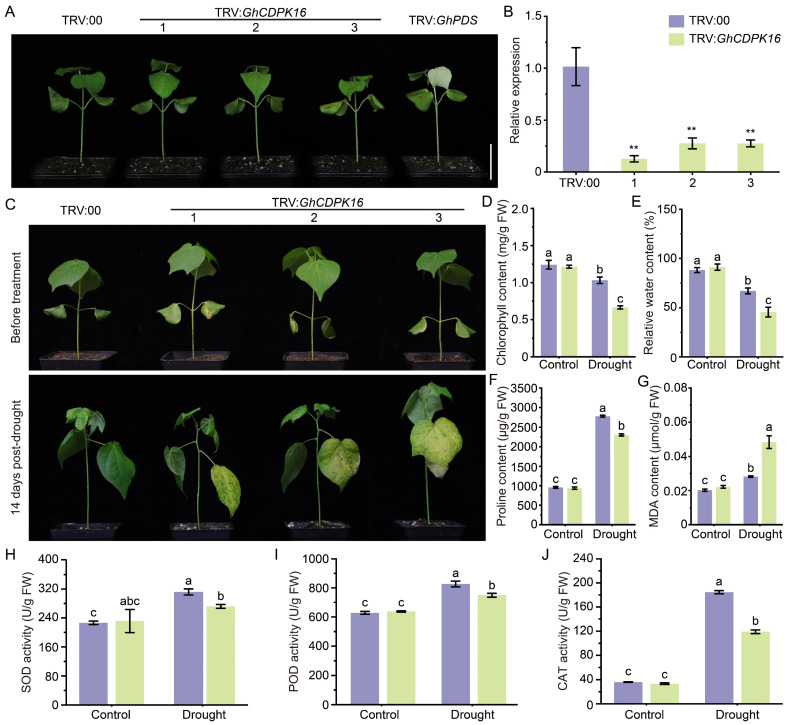
Silencing of *GhCDPK16* reduced drought tolerance in cotton. (**A**) Phenotypes of control and *GhCDPK16*-silenced plants via VIGS technology. The *GhPDS* gene was used as a marker, revealing an albino leaf phenotype following VIGS in cotton. Numbers 1 to 3 represented three distinct plants. (**B**) Relative expression levels of *GhCDPK16* in control and TRV:*GhCDPK16* plants. (**C**) Phenotypes of control and *GhCDPK16*-silenced plants before and after drought treatment. (**D**–**J**) The analysis of chlorophyll content (**D**), relative water content (**E**), proline content (**F**), MDA content (**G**), and activities of SOD (**H**), POD (**I**), and CAT (**J**) in control and *GhCDPK16*-silenced plants leaves after drought treatment. Values represent the mean ± SD from three biological replicates. ** *p* < 0.01 by Student’s *t* test. Columns with different letters indicate significant differences (*p* < 0.05, Student’s *t* test).

**Figure 7 ijms-25-08308-f007:**
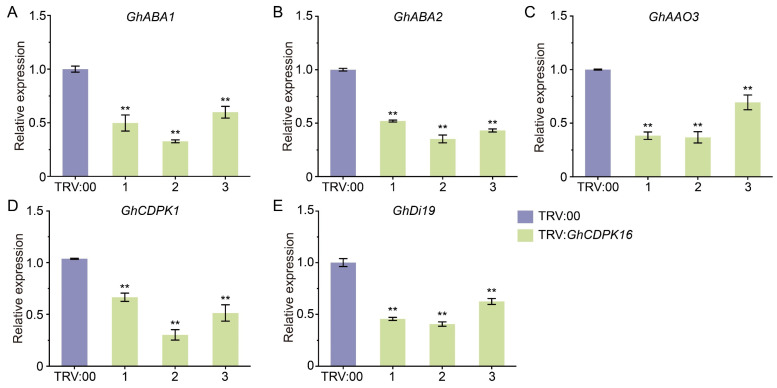
Expression levels of abiotic stress-related genes in control plants and *GhCDPK16*-silenced plants. (**A**–**E**) Relative expression levels of several ABA-related genes, including *GhABA1* (**A**), *GhABA2* (**B**), *GhAAO3* (**C**), *GhCDPK1* (**D**), and *GhDi19* (**E**) in control and *GhCDPK16*-silenced plants after drought treatment. Numbers 1 to 3 represented three distinct *GhCDPK16*-silenced plants. Values represent the mean ± SD from three biological replicates. ** *p* < 0.01 by Student’s *t* test.

## Data Availability

Data are contained within the article and Appendix A.

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
