# Peer review of "Calcium-Dependent Protein Kinase GhCDPK16 Exerts a Positive Regulatory Role in Enhancing Drought Tolerance in Cotton"

_ijms, 2024, doi:10.3390/ijms25158308_

Round 1

Reviewer 1 Report

Comments and Suggestions for Authors

The authors of the manuscript took up an interesting topic and conducted valuable research. The authors tried to explain the problem using the latest research techniques and based on physiological indicators, which is a major contribution to further research. The manuscript is well written, understandable and prepared in accordance with the requirements for scientific publication.

The Introduction explains in detail the subject of the research, based on information collected from well-selected scientific journals. In the Results, they skillfully described the obtained data and in the Discussion they made good references to the research of other authors.

Nevertheless, I have some comments regarding the description of the research performed in the materials and methods. I would suggest providing more detailed information that may be helpful to potential research recipients. Suggestions and comments below:

Line 387: Was the temperature the same day and night? what was the light intensity and type of lamps used (fluorescent or LED)

Line 407: what did cotton seedlings grow in? (soil, pot size)

Line 458: in what conditions the sown seeds germinated (light, day length, temperature. Size of the plates and amount of nutrient solution with different concentrations of mannitol.

Figure 5, Figure S2, Figure 7. Specify what plant the figures refers to.

Line 469: Leaves of what plant? (cotton or Arabidopsis) also Lines: 480, 488 and 503.

Comments on the Quality of English Language

Some sentences should be stylistically corrected

Author Response

We acknowledge your constructive comments and recommendations, which we found very helpful for improving our manuscript.

Comments 1 Line 387: Was the temperature the same day and night? what was the light intensity and type of lamps used (fluorescent or LED)

Response 1: Thank you for your careful comment. Following your suggestion, this description has been supplemented in our revised manuscript (Page 13, Line 395-398).

The corrected statement is below:

“They were grown in the growth chambers (PRG-I, Ningbo le electrical instrument manufacturing Co., LTD) with a constant temperature of 23°C, 70% humidity, LED lighting with a light intensity of 500 µmol/m²·s and a 16-hour light and 8-hour dark cycle.”

Comments 2 Line 407: what did cotton seedlings grow in? (soil, pot size)

Response 2: Thank you for your comment. Since the spatial and temporal expression analysis involves tissues from different developmental stages, the ‘TM-1’ used for sampling was grown in the field. Therefore, we have added this clarification in the revised manuscript (Page 14, Line 412-416).

The corrected statement is below:

“To investigate the spatial and temporal expression characteristics of GhCDPK16, we extracted RNA from various tissues of ‘TM-1’ grown in the field (Hangzhou, Zhejiang Province, China), including root, stem, leaf, cotyledon, apical bud, anther, filament, stigma, style, petal, sepal, torus, flower, as well as ovule and fiber at different developmental stages, respectively.”

Comments 3 Line 458: in what conditions the sown seeds germinated (light, day length, temperature. Size of the plates and amount of nutrient solution with different concentrations of mannitol.

Response 3: Thank you for your suggestion. All plants were germinated in the growth chamber as described in the 4.1 section of materials and methods (Page 13, Line 395-398). As for the size of the plates and amount of nutrient solution, we have supplemented detail information in our revised manuscript (Page 15, Line 460-462, Line 471-473, Line 481-483).

Comments 4 Figure 5, Figure S2, Figure 7. Specify what plant the figures refers to.

Response 4: Thank you for your careful comment. Following your suggestion, we have added detailed explanations in the figure legends of our revised manuscript (Page 8, Line 249-250; Page 9, Line 258-259; Page 12, Line 319-320).

Comments 5 Line 469: Leaves of what plant? (cotton or Arabidopsis) also Lines: 480, 488 and 503.

Response 5: Thank you for your careful comment. We have provided additional details in the Materials and Methods section of our revised manuscript (Page 15, Line 488-489, Line 500; Page 16, Line 509-510, Line 525-526).

Reviewer 2 Report

Comments and Suggestions for Authors

The article entitled " Calcium-dependent protein kinase GhCDPK16 exerts a positive regulatory role in enhancing drought tolerance in plants"   has tried to reveal the role of GhCDPK16 in drought tolerance in cotton. Some issues need to be concerned.

1. The title "....drought tolerance in plants",here "plants" may not suitable, may be "cotton" used accurately.

2. The authors only compare the GhCDPK16 and AtCDPK4,  AtCDPK11, how about the other copy of  GhCDPK16 in cotton, and phylogenetic tree ?

3. Why not the abiotic stress-related genes in GhCDPK60 overexpressed lines are different from these genes in GhCDPK16-silenced plants?

4. GhCDPK16 overexpressed lines whether affect the ABA biosynthesis pathways?

5. It's not clear GhCDPK16 positively enhanced drought tolerance in cotton via ABA signaling pathway, inducing closure of leaf stomata or other reasons?

6. The supplementary Figure S1 and S2 cannot be found.

Comments on the Quality of English Language

Minor editing of English language required.

Author Response

We acknowledge the reviewer’s constructive comments and recommendations, which we found very helpful for improving our manuscript.

Comments 1: The title "....drought tolerance in plants", here "plants" may not suitable, may be "cotton" used accurately.

Response 1: Thank you for pointing this out. We agree with your comment and have updated the term ‘plants’ to ‘cotton’ in the revised manuscript (Page 1, Line 4).

Comments 2: The authors only compare the GhCDPK16 and AtCDPK4, AtCDPK11, how about the other copy of GhCDPK16 in cotton, and phylogenetic tree?

Response 2: Thank you for your thorough feedback. The phylogenetic tree of GhCDPK family proteins was illustrated in Figure 1 of our previous study, “GhCDPK60 positively regulates drought stress tolerance in both transgenic Arabidopsis and cotton by regulating proline content and ROS level” [1]. As depicted in the Figure 1, GhCDPK16 exhibits a high degree of homology in protein sequences with AtCDPK4/AtCDPK11, which is why we concentrated on the functions of GhCDPK16 in this study.

Figure 1. Phylogenetic analysis of predicted CDPK proteins from three different plants species. The phylogenetic tree was generated from the alignment result of the full-length amino acid sequences by the neighbor-joining (NJ) method. All CDPKs members, together with homologs of Arabidopsis and rice, were classified into four distinct clades shown in different colors. The prefixes At, Os and Gh are used to identify CDPK proteins from A. thaliana, O. sativa and G. hirsutum, respectively.

Comments 3: Why not the abiotic stress-related genes in GhCDPK60 overexpressed lines are different from these genes in GhCDPK16-silenced plants?

Response 3: I assume you mean that the abiotic stress-related genes in the GhCDPK16 overexpressed lines differ from those in the GhCDPK16-silenced plants, but not from those in the GhCDPK60 overexpressed lines. In the GhCDPK16 overexpressed lines, we examined the expression levels of several transcription factors due to their broad regulatory effects on downstream genes under drought conditions. After confirming a significant increase in the expression of these transcription factors in the overexpression lines, we found that they were all related to the regulation of the ABA signaling pathway [2-6]. Therefore, we focused on the expression levels of genes associated with ABA biosynthesis in the GhCDPK16- silenced lines.

Comments 4: GhCDPK16 overexpressed lines whether affect the ABA biosynthesis pathways?

Response 4: Thank you for your comment. We observed that GhCDPK16 was significantly induced after ABA treatment, and the expression levels of some ABA signaling-related transcription factors were markedly increased in GhCDPK16-overexpressing Arabidopsis after drought treatment. This suggests that GhCDPK16 overexpressed lines may be involved in the ABA signaling pathway to some extent. Further experimental validation is needed, such as comparing the expression levels of ABA biosynthesis-related genes in the control and GhCDPK16-overexpressing lines after drought treatment, or assessing the sensitivity of these lines under different concentrations of ABA. This will require more time and may not be feasible within the specified five-day time frame. However, it is a crucial aspect that we will focus on in our future research. Thank you once again for your valuable suggestions.

Comments 5: It's not clear GhCDPK16 positively enhanced drought tolerance in cotton via ABA signaling pathway, inducing closure of leaf stomata or other reasons?

Response 5: Thank you for your valuable comment. It is well known that plants enhance drought tolerance through various mechanisms, including root development, osmotic regulation, antioxidant enzyme activity regulation, stomata regulation and ABA-mediated signaling pathways [7]. In this study, we found that GhCDPK16 actively regulates cotton drought tolerance by modulating root development, osmolyte proline content, and antioxidant enzyme activity. Additionally, the expression levels of ABA biosynthesis-related genes were significantly reduced in GhCDPK16-silenced lines. We speculated that the regulation of drought tolerance by GhCDPK16 may be related to the ABA signaling pathway, while stomata regulation and other factors warrant further investigation in future studies.

Comments 6. The supplementary Figure S1 and S2 cannot be found.

Response 6: Thank you for your comment. In our revised manuscript, Figure S1 and Figure S2 can be found on Line 124 of Page 4 and Line 254 of Page 9, respectively.

References:

  1. Yan, M.; Yu, X.; Zhou, G.; Sun, D.; Hu, Y.; Huang, C.; Zheng, Q.; Sun, N.; Wu, J.; Fu, Z.; Li, L.; Feng, Z.; Yu, S., GhCDPK60 positively regulates drought stress tolerance in both transgenic Arabidopsis and cotton by regulating proline content and ROS level. Front Plant Sci 2022, 13, 1072584.
  2. Qin, L. X.; Li, Y.; Li, D. D.; Xu, W. L.; Zheng, Y.; Li, X. B., Arabidopsis drought-induced protein Di19-3 participates in plant response to drought and high salinity stresses. Plant Mol Biol 2014, 86, (6), 609-25.
  3. Kim, J.-S.; Mizoi, J.; Yoshida, T.; Fujita, Y.; Nakajima, J.; Ohori, T.; Todaka, D.; Nakashima, K.; Hirayama, T.; Shinozaki, K.; Yamaguchi-Shinozaki, K., An ABRE Promoter Sequence is Involved in Osmotic Stress-Responsive Expression of the DREB2A Gene, Which Encodes a Transcription Factor Regulating Drought-Inducible Genes in Arabidopsis. Plant and Cell Physiology 2011, 52, (12), 2136-2146.
  4. Yokoi, S.; Quintero, F. J.; Cubero, B.; Ruiz, M. T.; Bressan, R. A.; Hasegawa, P. M.; Pardo, J. M., Differential expression and function of Arabidopsis thaliana NHX Na+/H+ antiporters in the salt stress response. The Plant Journal 2002, 30, (5), 529-539.
  5. Asif, M. A.; Zafar, Y.; Iqbal, J.; Iqbal, M. M.; Rashid, U.; Ali, G. M.; Arif, A.; Nazir, F., Enhanced expression of AtNHX1, in transgenic groundnut (Arachis hypogaea L.) improves salt and drought tolerence. Molecular biotechnology 2011, 49, (3), 250-6.
  6. Narusaka, Y.; Nakashima, K.; Shinwari, Z. K.; Sakuma, Y.; Furihata, T.; Abe, H.; Narusaka, M.; Shinozaki, K.; Yamaguchi‐Shinozaki, K., Interaction between two cis‐acting elements, ABRE and DRE, in ABA‐dependent expression of Arabidopsis rd29A gene in response to dehydration and high‐salinity stresses. The Plant Journal 2003, 34, (2), 137-148.
  7. Ullah, A.; Sun, H.; Yang, X.; Zhang, X., Drought coping strategies in cotton: increased crop per drop. Plant Biotechnol J 2017, 15, (3), 271-284.